# An Active Masked Attention framework for Many-to-Many Cross-Domain Recommendations

Feng Zhu
Ant Group
Hangzhou, China
zhufeng.zhu@antgroup.com

Xinxing Yang
Ant Group
Hangzhou, China
xinxing.yangxx@antgroup.com

Longfei Li
Ant Group
Hangzhou, China
longyao.llf@antgroup.com

Jun Zhou*
Ant Group
Hangzhou, China
jun.zhoujun@antgroup.com

## ABSTRACT

Cross-Domain Recommendation (CDR) has been proposed to improve the recommendation accuracy in the target domain (the sparser dataset) by benefiting from the auxiliary information transferred or the knowledge learned from one or many source domains (the denser datasets). However, most of the existing CDR approaches still suffer from the problem of negative transfer caused by undifferentiated knowledge transfer, and thus the recommendation accuracy in some domains, especially in the sparser domains, is still too low, which is not practical in real application scenarios. To address this problem, we propose a novel **A**ctive **M**asked **A**ttention framework, i.e., AMA-CDR, for many-to-many CDR scenarios. Our AMA-CDR pursues a higher goal for CDR approaches, i.e., *improving the recommendation performance in the target domain to achieve a practically usable level*, which is meaningful and challenging in real CDR systems. Specifically, AMA-CDR adopts an end-to-end graph embedding to reduce the objective distortion between graph embedding and embedding combination. More importantly, we propose an active mask for the embedding combination to ease negative transfer, which leverages both the prior knowledge, i.e., data density, and the posterior knowledge, i.e., sample uncertainty. Extensive experiments conducted on two public datasets demonstrate that our proposed AMA-CDR models significantly outperform the state-of-the-art approaches and achieve the new goal.

## CCS CONCEPTS

• **Information systems** → **Recommender systems**; • **Theory of computation** → *Active learning*; • **Computing methodologies** → Transfer learning.

## KEYWORDS

Cross-domain recommendation, active learning, graph embedding, and attention

*The corresponding author

**ACM Reference Format:**
Feng Zhu, Xinxing Yang, Longfei Li, and Jun Zhou. 2024. An Active Masked Attention framework for Many-to-Many Cross-Domain Recommendations. In *Proceedings of the 32nd ACM International Conference on Multimedia (MM '24), October 28-November 1, 2024, Melbourne, VIC, Australia.* ACM, New York, NY, USA, 10 pages. https://doi.org/10.1145/3664647.3681435

## 1 INTRODUCTION

### 1.1 Background

The data sparsity problem is a long-standing problem in many recommender systems, e.g., Amazon, Taobao, Facebook, and Tiktok, which may lead to over-fitting to the sparse training dataset. There are many classical graph-based and factorization-based solutions, e.g., Random Walk [7], Deep Walk [40], Probabilistic Matrix Factorization (PMF) [36], and Bayesian Personalised Ranking (BPR) [41], attempting to address this intractable problem in a single dataset. However, the training data in a sparse domain is very limited, which means there are still not enough well-trained embeddings of users/items for information propagation in the graph-based models and for neighbourhood integration in the factorization-based models. It is realised that without the help of the auxiliary information from other related domains, the traditional single-domain approaches cannot break the above-mentioned bottleneck. Thus, Cross-Domain Recommendation (CDR) [2] is proposed to leverage the auxiliary information from the dense domain to help improve the recommendation accuracy in the sparse domain.

### 1.2 Existing Solutions and Their Limitations

According to transfer paradigms, existing CDR models can be generally classified into four big groups: (1) one-to-one paradigm [4, 48, 54], (2) many-to-one paradigm [22, 39, 59, 60], (3) one-to-many paradigm [20, 22], and (4) many-to-many paradigm [8, 28, 37, 67]. In this paper, we mainly focus on the hardest but meaningful CDR scenario, i.e., many-to-many CDR, which leverages the knowledge from multiple domains (**many**) to improve the recommendation in multiple domains (**many**). The first two categories are called single-target CDR and the last two categories are called multi-target CDR as well in the literature [57, 66].

However, for the existing CDR approaches, the performance improvement in the target domain is still limited. Most of the existing approaches only focus on performance improvements compared with some single-domain models and their state-of-the-art CDR

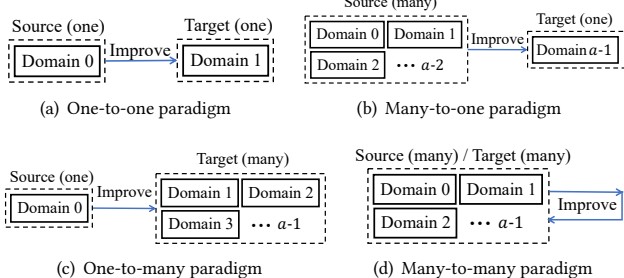

(a) One-to-one paradigm

(b) Many-to-one paradigm

(c) One-to-many paradigm

(d) Many-to-many paradigm

**Figure 1: Different transfer paradigms in CDR.**

models. For example, they can improve the sparse domains from 0.2 to 0.22 (i.e., the improvement is 10%) in terms of the NDCG (Normalised Discounted Cumulative Gain) metric. Although the performance improvement is significant compared with their poor baselines, the recommendation performance (NDCG = 0.2) is still too low in real application scenarios. It should be highlighted that if the performance in the target domains cannot reach an ideal level, the relative improvement of performance is useless in real recommendation scenarios. Therefore, in addition to enhancing the recommendation performance in all domains, in this paper, we also propose a new goal for CDR approaches, i.e., *improving the recommendation performance in the sparse domain to achieve a practically usable level, e.g., closing to the recommendation performance in the source domain.* (**New Goal**).

To achieve **New Goal**, we need to enhance the exchangeability of knowledge among multiple domains. In this paper, we construct interaction graphs for different domains and integrate these graphs via a suitable mask strategy, which can significantly enhance the exchangeability among multiple domains (as designed in Figure 2). This is because embedding propagation and combination for common users can mine the underlying relations among different graphs (domains). However, a research problem in the literature significantly reduces the effectiveness of the existing CDR approaches.

**Research Problem: Negative transfer caused by undifferentiated knowledge transfer.** With the increase in the number of auxiliary domains, if we cannot design a reasonable mechanism to punish the negative information from auxiliary domains, the problem of negative transfer seems to be inevitable (see the detailed problem definition and analysis in the supplementary material). Although some multi-task learning (MTL) approaches [31] and transfer learning (TL) approaches [6, 42, 53] have attempted to address this problem (see the approach details in Section 4.2), some transfer-related prior knowledge, e.g., data density and sample uncertainty, in the area of CDR are ignored in these approaches.

### 1.3 Our Approach and Contribution

To address the above problem and achieve **New Goal**, in this paper, we propose a novel Active Masked Attention framework for many-to-many CDR scenarios. The characteristics and contributions of our work are summarized as follows:

- In general, we design an end-to-end graph framework to reduce the objective distortion between graph embedding, i.e., our proposed revisiting graph embedding strategy, and transfer strategy, i.e., the attention for embedding combination.

- To address the problem of negative transfer in many-to-many CDR scenarios, we propose an active mask for the element-wise attention weights to ease the problem of negative transfer, which leverages both the prior knowledge, i.e., data density, and the posterior knowledge, i.e., sample uncertainty, to actively choose a suitable probability of uncertain samples. Actively selecting doubtful samples and giving them a higher probability to absorb the knowledge learned from other domains can reduce their uncertainty and thus enhance the performance of all sparser and denser domains.

- Extensive experiments conducted on two public datasets with seven domains demonstrate that our proposed AMA-CDR models significantly improve the state-of-the-art approaches by an average improvement of 25.34%. With the help of our AMA-CDR, in most experimental tasks, the sparser domain can even outperform the denser domain.

## 2 THE PROPOSED MODEL

The research problem in Section 1, i.e., negative transfer, motivates us to design an effective end-to-end framework for many-to-many CDR scenarios. In this section, we introduce the general structure of our AMA-CDR and explain the rationale behind the structure. After that, we present the detailed components of our AMA-CDR, i.e., a novel **A**ctive **M**asked **A**ttention framework for many-to-many CDR scenarios. Due to the space limitations, we explain the main notations and the training samples in the supplementary material.

### 2.1 Graph Embedding

For each domain in many-to-many CDR scenarios, the users and items can form a heterogeneous graph via their interaction relations. Graph embedding has shown its powerful ability in embedding generation for recommendation [11, 30, 49, 65]. Therefore, we tend to use a graph embedding method to respectively generate the embeddings of users and items in each domain, e.g., $U^x$ and $V^x$ in the domain $D^x$ (see the graph embedding of Figure 2).

To design an effective end-to-end framework, the structure of graph embedding should be lightweight rather than complex. Therefore, in this paper, we only apply a revisited graph neural network to learn the embeddings of users/items. Next, we will introduce the process of linear embedding propagation.

*2.1.1 Linear embedding propagation.* Given ratings of users to items, the user-item bipartite graph can be denoted as $G = (\{\mathcal{U}, \mathcal{V}\}, E)$, with $E$ is the set of user-item interaction relationships, i.e., observed ratings. Inspired by the revisiting graph-based collaborative filtering in [3], the updating rule of user/item embeddings can be represented as follows.

$$U_i^{x,q+1} = [\frac{1}{d_i} U_i^{x,q} + \sum_{j \in R_i} \frac{1}{\sqrt{d_i \times d_j}} V_j^{x,q}]W^{x,q},$$
$$V_j^{x,q+1} = [\frac{1}{d_j} V_j^{x,q} + \sum_{i \in R_j} \frac{1}{\sqrt{d_j \times d_i}} U_i^{x,q}]W^{x,q}, \tag{1}$$

where $U_i^{x,q+1}$ is the embedding of user $u_i$ in domain $D^x$ at the iteration step $q + 1$, $d_i$ or $d_j$ is the diagonal degree of user $u_i$ or item $v_j$ in graph $G^x$, $V_j^{x,q+1}$ is the embedding of item $v_j$ in domain $D^x$ at the iteration step $q + 1$, $R_i$ or $R_j$ is the neighbor set of the node

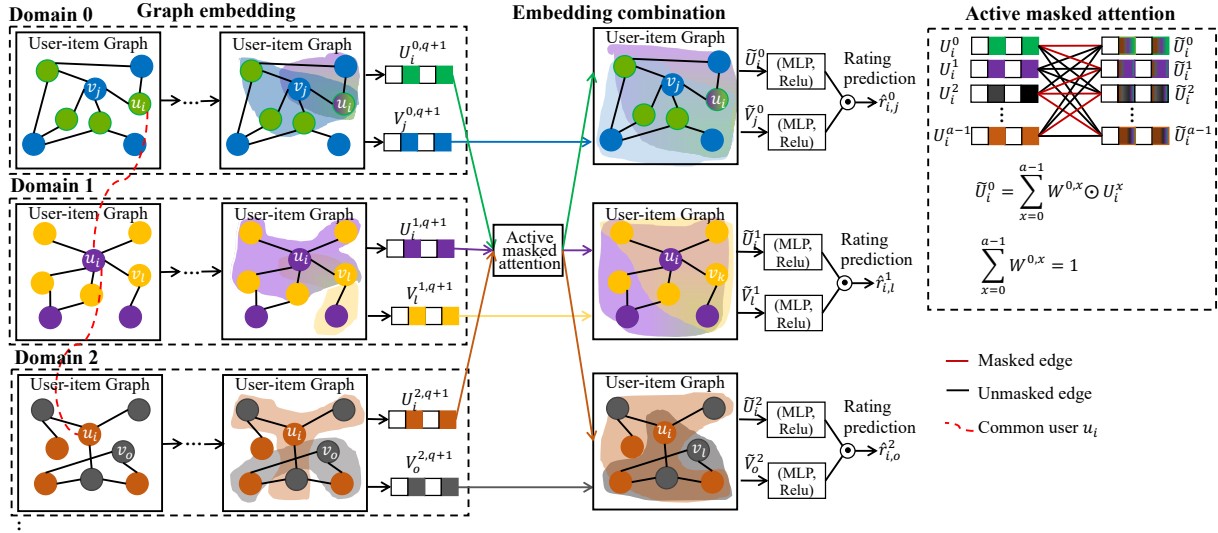

**Figure 2: The structure of our proposed AMA-CDR model.**

(user $u_i$ or item $v_j$) in graph $G^x$, and $W^{x,q}$ is the transformation parameters for domain $D^x$.

## 2.2 Embedding Combination

Traditional transfer strategies tend to transfer the knowledge learned from one domain to another domain. However, in many-to-many CDR scenarios, things become more complex as we need to leverage the data of multiple domains (many) to improve the recommendation performance themselves (many). Unlike the traditional transfer strategies, in this paper, we combine the embeddings of common entities generated from multiple domains, i.e., their corresponding user-item graphs. By doing so, the combined embeddings of common entities for each domain can retain all knowledge learned from the multiple domains to different extents. To this end, the fine-grained attention mechanism has been widely used in the literature [26, 27, 62, 67].

For a common entity (a user $u_i$ or an item $v_i$), our attention mechanism pays different attention to the user/item embeddings generated from different domains. The biggest advantage of this mechanism is that for an individual common entity, we can integrate their representations from different domains in different proportions. Therefore, it is expected to reduce the effect of inaccurate embeddings learned from sparser domains and emphasize the accurate embeddings learned from denser domains. Also, the attention mechanism is fine-grained that corresponds to the level of elements, and thus remains more informative elements (latent representation) from each set of embedding elements in $\{U_i^0, U_i^1, ..., U_i^{a-1}\}$ or $\{V_j^0, V_j^1, ..., V_j^{a-1}\}$. The combined embedding $\tilde{U}_i^x$ of a common user $u_i$ in the domain $D^x$ can be represented as:

$$\tilde{U}_i^x = \sum_{y=0}^{a-1} W^{x,y} \odot U_i^y, \quad \sum_{y=0}^{a-1} W^{x,y} = <1, 1, ..., 1> \in \mathbb{R}^k, \quad (2)$$

where $W^{x,y} \in \mathbb{R}^k$ is the weight vector of domain $x$ for domain $y$ and $\odot$ denotes the element-wise multiplication, and $k$ is the embedding dimension of $U_i^y$. In fact, we apply the softmax activation

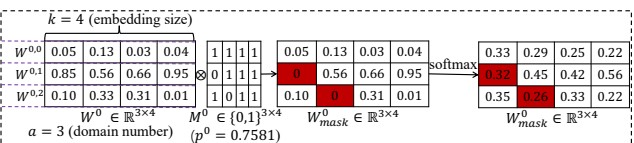

**Figure 3: The masking process.**

function to guarantee $\sum_{y=0}^{a-1} W^{x,y} = \mathbf{1}^k$. Similarly, we can obtain the combined embedding $\tilde{V}_j^x$ of a common item $v_j$ in the domain $D^x$ if there are common items among the multiple domains.

## 2.3 Uncertainty and Active Mask

As introduced in Section 2.2, the attention mechanism can control the importance of embeddings of users/items learned from multiple domains according to its attention weights for different domains. However, the attention weights are trained via the training samples in different domains, which means that the attention weights for sparser domains are hard to be properly tuned because of limited training samples. This is a major cause of negative transfer in many-to-many CDR scenarios. Accordingly, in this section, we propose another promising solution, i.e., a new active mask on the attention mechanism, to achieve a better recommendation performance in many-to-many CDR scenarios. Especially for sparser domains, with the help of the proposed active mask, we can verify the claim in Section 1, i.e., sparser domains can outperform denser domains.

*2.3.1 Underlying rationale.* The proposed attention mechanism in Section 2.2 ignores the difference among multiple domains and expects that the attention weights can be properly tuned. To avoid negative transfer, we can obtain some useful **prior knowledge** about the difference among multiple domains according to the statistics of training datasets. In this paper, we choose the density of training datasets to measure the demand degree of transferred knowledge from other domains. The general idea of this strategy is that if a domain is sparse, then it needs more transferred knowledge from

other domains, and vice versa. Specifically, we will mask fewer attention weights for sparser domains (transferring more knowledge from other domains) and mask more attention weights for denser domains (remaining more knowledge learned from themselves). A similar strategy has been used in some existing CDR approaches, e.g., DCDCSR[64], which leverages the sparsity degrees to combine the user embeddings learned from different domains. However, the density of training datasets is general knowledge to measure the difference among different domains, and we still cannot distinguish the difference among different training samples. In addition to the generally masked probabilities for multiple domains according to their density, inspired by the basic idea of active learning, we also need to give more opportunities for the attention weights of uncertain samples in each training epoch to not be masked. This means that we need to transfer more knowledge from other domains for uncertain samples rather than for certain samples in each training epoch. Note that the uncertainty is a **posterior knowledge** that needs to be computed in each training batch. Therefore, in this section, we will utilise both the prior knowledge, i.e., density, and the posterior knowledge, i.e., uncertainty, to control the demand degree of transferred knowledge in the process of embedding combination.

*2.3.2 Demand probability of transferred knowledge.* According to the density, the demand probability $p^x$ of transferred knowledge of domain $D^x$ can be represented as follows.

$$p^x = \frac{(den^t - den^x)}{den^t}, \quad den^t = \sum_{x=1}^{a-1} den^x, \quad (3)$$

where $den^x$ is the density of domain $D^x$ and $den^t$ is the total density of all domains. Here, the demand probability $p^x$ is a relative probability compared with other domains. For example, as shown in Table 1 of Section 3, in the Douban datasets, we can obtain the demand probabilities of all the three domains $p^0 = 0.7581$ (Book domain), $p^1 = 0.7292$ (Music domain), and $p^2 = 0.5126$ (Movie domain), according to Eq. (3).

*2.3.3 Active masked attention.* In this section, to avoid negative transfer, we set a mask on the attention weights of the domain $D^x$ according to the demand degree of transferred knowledge, i.e. $p^x$, from other domains. For domain $D^x$, its attention weights for all domains (including itself) can be denoted as $W^x = \{W^{x,0}, W^{x,1}, ..., W^{x,a-1}\}$, $W^x \in \mathbb{R}^{a \times k}$. For clarity, we take an example to show the masking process in Figure 3. Note that for a target domain $D^x$, we do not mask the attention weights of $D^x$ for $D^x$. This means that a target domain $D^x$ always retains the knowledge learned from itself and absorbs the knowledge learned from other domains according to $p^x$. After many training iterations, our proposed AMA-CDR can mask fewer attention weights for sparser domains (transferring more knowledge from other domains) and mask more attention weights for denser domains (remaining more knowledge learned from themselves).

As analysed in Section 2.3.1, although the demand degree of transferred knowledge is the general knowledge to measure the difference among different domains, we still cannot distinguish the difference among different training samples. To address this problem, in each training step, we first forward to obtain the uncertainties of a batch of training samples, and then mask the attention

---

**Algorithm 1:** Active masking function $AM_1$

**Input:** The attention weights of domain $D^x$
    $W^x = \{W^{x,0}, W^{x,1}, ..., W^{x,a-1}\}$, the demand probability
    $p^x$ of domain $D^x$ (the prior knowledge)

**Output:** Masked weights $W^x_{mask}$

1   $P_{active} \leftarrow \emptyset$;
   /* Uncertainty prediction stage: calculate the
    uncertainties (active mask probabilities)      */
2   $stage \leftarrow$ 'uncertainty prediction' ;
3   $W^x_{no-mask} \leftarrow AM_2(W^x, p^x, stage, P_{active})$;     /* forward to
    obtain $W^x$ without a mask, $AM_2$ is the detailed
    active masking function (see Algorithm 2) */
4   Obtain the combined embeddings $\tilde{U}^x$ and $\tilde{V}^x$ according to Eq. (2);
5   Obtain the predicted ratings according to Eqs. (6 and 7) ;
6   Calculate the uncertainties $\gamma$ of all samples in this training batch
    according to Eq. (4);
7   $P_{active} = normalise(\gamma)$;
   /* Active masking stage: Call $AM_2$ function again and
    obtain masked weights                  */
8   $stage \leftarrow$ 'active masking' ;
9   $W^x_{mask} \leftarrow AM_2(W^x, p^x, stage, P_{active})$;
10   **return** $W^x_{mask}$

---

**Algorithm 2:** Active masking function $AM_2$

**Input:** The attention weights of the domain $D^x$
    $W^x = \{W^{x,0}, W^{x,1}, ..., W^{x,a-1}\}$, the demand probability
    $p^x$ of the domain $D^x$ (the prior knowledge), the stage
    $stage$, and the mask probabilities $P_{active}$

**Output:** Masked weights $W^x_{mask}$

1   $W^x_{mask} \leftarrow \emptyset$;
2   **for** $W^{x,j}$ *in* $W^x$ **do**
3      $W^{x,j}_{mask} = W^{x,j}$;
4      **if** *the mode is training and* $stage ==$ *'active masking'* **then**
5          **if** $x \,!= j$ **then**
6              $mask \leftarrow p_{active} < p^x$;
7              $W^{x,j}_{mask} = W^{x,j} \otimes mask$;
8      Append $W^{x,j}_{mask}$ to $W^x_{mask}$;
9   **end**
10   $W^x_{mask} = softmax(W^x_{mask})$;
11   **return** $W^x_{mask}$

---

weights for uncertain samples. This is because uncertain samples lack confidence in their predictions, and thus they are more likely negatively affected by the knowledge transferred from other domains. In this paper, we use the entropy of the prediction to measure the uncertainty. The larger is the entropy, the more uncertain is the prediction. The uncertainty $\gamma$ of a sample can be represented as follow.

$$\gamma = -\hat{y} * log(\hat{y}) - (1 - \hat{y}) * log(1 - \hat{y}). \quad (4)$$

The detailed processes are demonstrated in Algorithms 1 and 2. We can obtain the masked attention weights $W^x_{mask}$ for domain $D^x$ according to Algorithms 1 and 2. Therefore, the function of the embedding combination (Eq. (2)) will be improved as follows.

$$\tilde{U}^x_i = \sum_{y=1}^{a-1} W^{x,y}_{mask} \odot U^y_i, \quad \sum_{y=1}^{a-1} W^{x,y}_{mask} = \mathbf{1}^k = <1, 1, ..., 1> \in \mathbb{R}^k. \quad (5)$$

**Table 1: Experimental datasets and tasks**

| Datasets | | Douban | | | Amazon | | |
|---|---|---|---|---|---|---|---|
| Domains | Book | Music | Movie | Music | Electronics | Movie | Video Games |
| #Users | 2,110 | 1,672 | 2,712 | 4,838 | 51,397 | 40,007 | 6,683 |
| #Items | 6,777 | 5,567 | 34,893 | 8,367 | 79,626 | 48,770 | 14,750 |
| #Interactions | 96,041 | 69,709 | 1,278,401 | 59,264 | 838,179 | 1,135,485 | 127,575 |
| Density | 0.67% | 0.75% | 1.35% | 0.15% | 0.02% | 0.06% | 0.13% |
| Demand probability $p$ | 0.7581 | 0.7292 | 0.5126 | 0.5833 | 0.9444 | 0.8333 | 0.6389 |

| Tasks | | Sparser | Denser | Overlap |
|---|---|---|---|---|
| Two-to-two CDR | Task 1 | DoubanBook | DoubanMovie | #Common Users = 2,106 |
| | Task 2 | DoubanMusic | DoubanMovie | #Common Users = 1,666 |
| | Task 3 | AmazonMusic | AmazonElectronics | #Common Users = 544 |
| | Task 4 | AmazonMovie | AmazonVideoGames | #Common Users = 1,629 |

| Tasks | | Domains | Overlap |
|---|---|---|---|
| Many-to-many CDR | Task 5 | DoubanBook+DoubanMusic+DoubanMovie | #Common Users = 1,662 |
| | Task 6 | AmazonMusic+AmazonElectronics+AmazonMovie+AmazonVideoGames | #Common Users = 75 |

## 2.4 Model Training and Prediction

As introduced in Section 2.3, we can obtain the masked attention weights for the combined embeddings of users and items, i.e., $\tilde{U}^x$ and $\tilde{V}^x$, for domain $D^x$. As shown in Figure 2, we employ an MLP to learn the interaction relations between users and items. Therefore, we can obtain the output of the MLP in the domain $D^x$ as follows.

$$U_{out}^x = f_u(\tilde{U}^x, \Theta_{MLP,U}^x), \quad V_{out}^x = f_v(\tilde{V}^x, \Theta_{MLP,V}^x), \quad (6)$$

where $f_u(*)$ and $f_v(*)$ are the MLP networks with the activation function *ReLU* for user and item embeddings. $\Theta_{MLP,U}^x$ and $\Theta_{MLP,V}^x$ are the corresponding parameters of user MLP and item MLP.

Next, we can obtain the rating predictions in the domain $D^x$ via a cosine similarity as follows:

$$\hat{Y}^x = cosine(U_{out}^x, V_{out}^x) = \frac{U_{out}^x \cdot V_{out}^x}{\|U_{out}^x\| \|V_{out}^x\|}. \quad (7)$$

To avoid the seesaw phenomenon caused by a joint objective, we train our proposed model in each domain via a separate optimiser. The objective function of our proposed AMA-CDR in domain $D^x$ can be represented as follows.

$$\min_{U_{out}^x, V_{out}^x, \Theta^x} \sum_{y \in Y^{x+} \cup Y_s^{x-}} \ell(\hat{y}, y) + \lambda(\|U_{out}^x\|_F^2 + \|V_{out}^x\|_F^2). \quad (8)$$

## 3 EXPERIMENTS AND ANALYSIS

We conduct extensive experiments on two real-world datasets, i.e., the Douban dataset with three domains and the Amazon dataset with four domains, to answer the following key questions:

- **Q1:** How does our AMA-CDR model perform when compared with the state-of-the-art models (see Result 1)?
- **Q2:** Can sparser domains outperform denser domains with the help of our AMA-CDR (see Result 1)?
- **Q3:** How do the end-to-end attention and the active mask contribute to performance improvement (see Result 2)?
- **Q4:** How does our AMA-CDR model perform when the negative transfer occurs (see Result 3)?

## 3.1 Experimental Setting

*3.1.1 Experimental Datasets and Tasks.* To validate the performance of our AMA-CDR and the state-of-the-art (SOTA) baselines, we choose two public datasets, i.e., Douban[1] [63] with the

three domains (DoubanBook, DoubanMusic, and DoubanMovie) and Amazon[2] [21] with the four domains (AmazonMusic, AmazonElectronics, AmazonMovie, and AmazonVideoGames). For the Douban dataset, we retain the users and items with at least 5 interactions, while for the Amazon dataset, we retain the users and items with at least 10 interactions. The same filtering strategy has been widely used in the existing approaches [56, 67]. Also, to train the CDR models smoothly, we serialise the user IDs of the multiple domains. The common users from different domains have duplicate IDs and distinct users in different domains have different IDs. Therefore, all users from different domains are in the same user space. For example, in the three domains of the Douban dataset, all the users from these three domains are mapped into the same user space. In this way, we can easy to combine the embeddings of common users from different domains. Note that for the Douban dataset and the Amazon dataset, the items are different among different domains, and thus we cannot combine the embeddings of common items. We list the dataset statistics and tasks in Table 1.

*3.1.2 Parameter Setting.* To ensure a fair comparison, we optimize the hyper-parameters of our AMA-CDR with those SOTA baseline models according to the parameter settings reported in their original papers. For all experimental models, we set the maximum number of training epochs to 50 and only report the best-performing result of all training epochs. For our proposed AMA-CDR, we randomly choose 7 unobserved interactions for each observed interaction (positive sample) as negative samples, set the depth of the revisiting graph $q$ (in Eq. (1)) as 2, the dimension $k$ of user/item embeddings is 32, the structure of the MLP for user embeddings and item embeddings in Eq. (6) is '$k$', the regularisation coefficient $\lambda$ is 0.001, the learning rate is 0.01, the optimiser function is Adam [19], and the batch size is 1,024. We list the demand probability of transferred knowledge $p$ for each domain in Table 1.

*3.1.3 Evaluation Metrics.* Similar to the majority of SOTA baselines [12, 67], we also adopt the *leave-one-out evaluation*. This evaluation strategy will scan all users in a domain, and for each test user, we choose his/her latest interaction as a positive testing sample and randomly select 99 observed interactions for this test user as negative testing samples. The process of evaluation is to rank these 100 testing samples according to the prediction results of recommendation models and observe the positive testing samples in the ranking list. There are two metrics for this evaluation strategy, i.e., *Hit Ratio (HR)* and *Normalised Discounted Cumulative Gain (NDCG)* [51]. *HR@N* means that the recall rate when the positive testing sample rank in the top-*N* list. In contrast, *NDCG@N* measures the ranking quality of the positive testing sample in the top-*N* list.

*3.1.4 Comparison Methods.* Considering the comparison fairness, our proposed AMA-CDR can only be compared with single-domain recommendation baselines in each domain, two-to-two CDR baselines in both two domains (a specific case of the many-to-many transfer paradigm), and many-to-many CDR baselines in multiple domains. Thus, we choose seven SOTA or representative baselines in three groups, i.e., (1) single-domain recommendation (SDR), (2) two-to-two CDR, and (3) many-to-many CDR. To validate the detailed contribution of the main components of our proposed

---

[1]Douban dataset URL: https://github.com/FengZhu-Joey/GA-DTCDR/tree/main/Data

[2]Amazon dataset URL: http://jmcauley.ucsd.edu/data/amazon/

**Table 2: The comparison of the baselines and our methods**

| | Model | | Training labels | Encoding | Embedding | Training paradigm | Optimization target | Transfer strategy |
|---|---|---|---|---|---|---|---|---|
| Baselines | Single-domain recommendation (SDR) | **NeuCF** [12] | Interaction label (0/1) | One-hot | Non-linear MLP | End-to-end | Single target | - |
| | | **LightGCN** [11] | Interaction label (0/1) | One-hot | GCN | End-to-end | Single target | - |
| | Two-to-two CDR | **DDTCDR** [24] | Rating | One-hot & multi-hot | Non-linear MLP | End-to-end | Dual targets (joint loss) | Dual transfer learning |
| | | **Bi-TGCF** [30] | Interaction label (0/1) | Heterogeneous graph | Graph embedding | End-to-end | Dual targets (joint loss) | Combination (concatenation/ average pooling) |
| | Many-to-many CDR | **MMOE** [32] | Rating | One-hot | Non-linear MLP | End-to-end | Multiple targets (joint loss) | Multi-task learning & mixture of experts |
| | | **GA-MTCDR-P** [67] | Rating | Heterogeneous graph | Graph embedding | Two-steps | Multiple targets (multiple separated losses) | Element-wise attention & personalization |
| | | **NMCDR** [55] | Rating | Heterogeneous graph | Graph embedding | End-to-end | Multiple targets (joint loss) | Node matching |
| Our Methods | Many-to-many CDR | **AMA-CDR-A** (a variant of AMA-CDR) | Rating | Heterogeneous graph | Revisiting GNN | End-to-end | Multiple targets (multiple separated losses) | Element-wise attention |
| | | **AMA-CDR-RM** (a variant of AMA-CDR) | Rating | Heterogeneous graph | Revisiting GNN | End-to-end | Multiple targets (multiple separated losses) | Element-wise attention & random masking |
| | | **AMA-CDR** (full-version) | Rating | Heterogeneous graph | Revisiting GNN | End-to-end | Multiple targets (multiple separated losses) | Element-wise attention & active masking |

**Table 3: The experimental results (HR@10 & NDCG@10) for Tasks 1 to 6 (the results of best-performing baselines are marked with '*' while the results of our best-performing models are marked with bold font). Note that the SDR models can be applied in each domain and thus we can obtain their results on seven domains (3 Douban domains and 4 Amazon domains). The two-to-two CDR models can not be applied in many-to-many tasks, i.e., Tasks 5 and 6, but the many-to-many CDR models can be applied in two-to-two tasks, i.e., Tasks 1 to 4**

| Task | Domain (D: Denser S: Sparser) | SDR baselines NeuCF HR NDCG | SDR baselines LightGCN HR NDCG | Two-to-two CDR baselines DDTCDR HR NDCG | Two-to-two CDR baselines Bi-TGCF HR NDCG | Many-to-many CDR baselines MMOE HR NDCG | Many-to-many CDR baselines GA-MTCDR-P HR NDCG | Many-to-many CDR baselines NMCDR HR NDCG | Our Method AMA-CDR HR NDCG | Improvement (AMA-CDR vs. best baselines) HR NDCG | Can sparser domain achieve the new goal? |
|---|---|---|---|---|---|---|---|---|---|---|---|
| Task 1 | DoubanBook (S) | .3899 .2182 | .3956 .2264 | .4242 .2562 | .4412 .2637 | .4342 .2591 | .4771* .2899* | .4511 .2664 | **.6995 .4425** | 46.61%↑ 45.74%↑ | Yes |
| | DoubanMovie (D) | .5411 .2991 | .5652 .3241 | .5666 .3272 | .6068 .3586 | .6723 .4211 | .6742* .4277* | .6023 .4116 | **.6812 .4308** | 1.03%↑ 0.73%↑ | |
| Task 2 | DoubanMusic (S) | .3198 .1771 | .3312 .2031 | .3493 .1892 | .4288* .2233 | .4244 .2412 | .4174 .2452* | .4119 .2318 | **.6920 .4542** | 67.27%↑ 85.95%↑ | Yes |
| | DoubanMovie (D) | .5411 .2991 | .5652 .3241 | .5681 .3307 | .6064 .3716 | .6322 .3978 | .6672* .4121* | .6011 .4066 | **.6742 .4183** | 1.05%↑ 1.50%↑ | |
| Task 3 | AmazonElectronics (S) | .4041 .2301 | .4066 .2465 | .4235 .2549 | .6447 .4385 | .7112 .4555 | .7741* .4920 | .7165 .5027* | **.8178 .5341** | 8.85%↑ 6.25%↑ | Yes |
| | AmazonMusic (D) | .3834 .2156 | .3906 .2169 | .4070 .2194 | .5823 .3801 | .6021 .3566 | .6144* .3755* | .6055 .3690 | **.7526 .5146** | 25.05%↑ 39.84%↑ | |
| Task 4 | AmazonMovie (S) | .5772 .3502 | .5900 .3670 | .6051 .3737 | .7805* .5475* | .7201 .4991 | .7502 .5144 | .7422 .5075 | **.8053 .5476** | 3.17%↑ 0.02%↑ | No |
| | AmazonVideoGames (D) | .4507 .2751 | .4524 .2831 | .4700 .2587 | .7507 .5187 | .7311 .5029 | .7557* .5118* | .7397 .4996 | **.8109 .5726** | 6.65%↑ 11.88%↑ | |
| Task 5 | DoubanBook (S) | .3899 .2182 | .3956 .2264 | - | - | .4611 .2881 | .4805* .3084 | .4799 .3339* | **.7056 .4531** | 46.85%↑ 35.69%↑ | Yes |
| | DoubanMusic (S) | .3198 .1771 | .3312 .2031 | - | - | .3922 .2289 | .4165 .2449 | .5091* .3370* | **.6967 .4554** | 36.85%↑ 35.13%↑ | |
| | DoubanMovie (D) | .5411 .2991 | .5652 .3241 | - | - | .6229 .3811 | .6761* .4280* | .6429 .4008 | **.6938 .4370** | 2.62%↑ 2.10%↑ | |
| Task 6 | AmaoznElectronics (S) | .4041 .2301 | .4066 .2465 | - | - | .6799 .4213 | .7649* .4869 | .7306 .4990* | **.8226 .5459** | 7.54%↑ 9.40%↑ | Yes |
| | AmazonMovie (S) | .5772 .3502 | .5900 .3670 | - | - | .6911 .4341 | .7614* .5222* | .7249 .5195 | **.8213 .5793** | 7.87%↑ 10.93%↑ | |
| | AmazonVideoGames (S) | .4507 .2751 | .4524 .2831 | - | - | .6933 .4415 | .7600* .5218* | .7461 .5144 | **.8406 .5760** | 10.61%↑ 10.39%↑ | |
| | AmazonMusic (D) | .3834 .2156 | .3906 .2169 | - | - | .6111 .3910 | .6286 .3809 | .6623* .4671* | **.7972 .5544** | 20.37%↑ 18.69%↑ | |

AMA-CDR framework, i.e., random masked attention and active masked attention, we conduct an ablation study to compare the two variants, i.e., AMA-CDR-A (only with end-to-end element-wise attention) and AMA-CDR-RM (with random masked attention), with the full-version AMA-CDR (with active masked attention). For a clear comparison, in Table 2, we list the technical details of all experimental models, including training labels, encoding, embedding, training paradigm, optimization target, and transfer strategies.

## 3.2 Result 1: Performance Comparison and Analysis (for Q1 and Q2)

To answer **Q1**, we compare the recommendation performance of our AMA-CDR with those of the seven baselines listed in Table 2. We report their experimental results in Table 3. Table 3 demonstrates the experimental results in terms of HR@10 & NDCG@10 for the six experimental tasks (see the tasks in Table 1). As we can

see from Table 3, our proposed AMA-CDR can outperform all the SDR, two-to-two CDR, and many-to-many CDR baselines by an average improvement of 30.60%, including HR@10 & NDCG@10. In particular, our AMA-CDR improves the best-performing baselines (with results marked by * in Table 3) by an average of 30.14% for Task 1, an average of 39.42% for Task 2, an average of 22.46% for Task 3, an average of 6.21% for Task 4, an average of 29.49% for Task 5, and an average of 11.47% for Task 6. Also, for the sparser domain, our AMA-CDR can significantly improve the performance of the best-performing baseline, and the performance of the sparser domain can reach and even surpass that of the denser domain (for **Q2**). Therefore, for **Q2**, our AMA-CDR can help the sparser domain outperform the denser domain. Note that we mark the detailed results for **Q2** in the last column of Table 3. In addition to Task 4, the performance of the sparser domains in other tasks can outperform those of the denser domains.

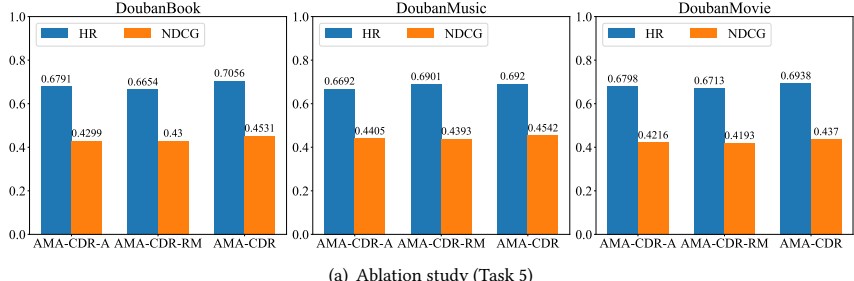

(a)  Ablation study (Task 5)

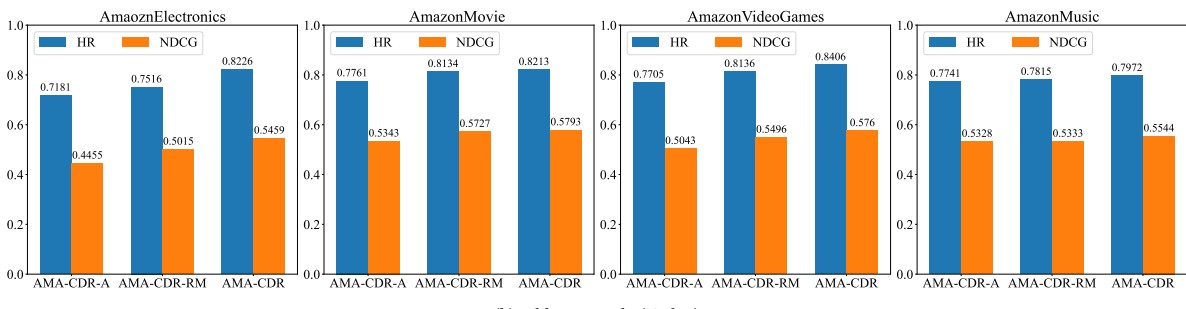

(b)  Ablation study (Task 6)

**Figure 4: Ablation study in tasks 5 and 6. AMA-CDR-A (revisiting GNN + element-wise attention) and AMA-CDR-RM (revisiting GNN + element-wise attention + random masking) are the two variants of our proposed AMA-CDR.**

**Comparison analysis.** As for **Q1**, our proposed AMA-CDR can outperform the SOTA or representative baselines due to (1) the end-to-end training paradigm that can reduce the objective distortion between the generation of graph embeddings and transfer process, and (2) the active mask mechanism that can leverage both the prior knowledge (data density) and the posterior knowledge (uncertainty of training samples) to avoid the problem of negative transfer. As for **Q2**, the performance of the sparser domain can reach and even surpass that of the denser domain because a higher demand probability $p$ (see Table 1) of transferred knowledge from other domains can force the recommendation system in the sparser domain to absorb more knowledge (user/item embeddings) from other domains rather than to retain their own inaccurate knowledge.

### 3.3  Result 2: Ablation Study (for Q3)

To answer **Q3**, we implement two variants of our proposed AMA-CDR, i.e., AMA-CDR-A (only with element-wise attention and without any mask strategies) and AMA-CDR-RM (with random masked attention). To validate the detailed contributions of the two components of our proposed AMA-CDR, i.e., random masked attention and active masked attention, we compare the two variants with our full-version AMA-CDR and report their results for Tasks 5 and 6 in Figure 4.

Firstly, as shown in Figure 4, even only with revisiting GNN and element-wise attention, our end-to-end AMA-CDR-A can significantly improve the SOTA or representative baselines in terms of HR@10 & NDCG@10. This means that if we can design suitable graph embedding and transfer strategies, our end-to-end models can significantly ease the problem of objective distortion.

Secondly, comparing AMA-CDR-RM with AMA-CDR-A, we can see from Figure 4 that with the help of the random mask, i.e., leveraging the prior knowledge – data density, our AMA-CDR-RM can ease

the problem of negative transfer to some extent. This means that the prior knowledge, i.e., data density, can guide our CDR model to effectively combine the knowledge, i.e., user/item embeddings, from other domains.

Finally, comparing AMA-CDR (the full version) with AMA-CDR-RM, we can see from Figure 4 that with the help of the active mask, i.e., leveraging both the prior knowledge – the data density and the posterior knowledge – the uncertainty of training sample, our AMA-CDR can further address the problem of negative transfer. Actively selecting uncertain samples and giving them a higher probability to absorb the knowledge learned from other domains can reduce their uncertainty and thus enhance the performance of all sparser and denser domains.

### 3.4  Result 3: Negative Transfer (for Q4)

To validate the performance of our AMA-CDR when the negative transfer occurs, we report the performance of the best-performing baseline, i.e., GA-MTCDR-P, and our AMA-CDR, and compare the changes from Task 1/2 to Task 5 (Douban dataset) and from Task 3/4 to Task 6 (Amazon dataset). For the Douban dataset, from Task 1/2 (Book+Movie or Music+Movie) to Task 5 (Book+Music+Movie), we only add a new domain, while for the Amazon dataset. from Task 3/4 (Music+Electronics or Movie+VideoGames) to Task 6 (Music+Electronics+Movie+VideoGames), we add two new domains. As we can see from Figure 5, from Task 1/2 to Task 5 and from Task 3/4 to Task 6, our AMA-CDR can improve the performance in all domains while GA-MTCDR-P occurs the phenomenon of negative transfer, i.e., with more auxiliary domains, the performance of GA-MTCDR-P decreases in some cases. This result demonstrates that our AMA-CDR can avoid negative transfer to a large extent.

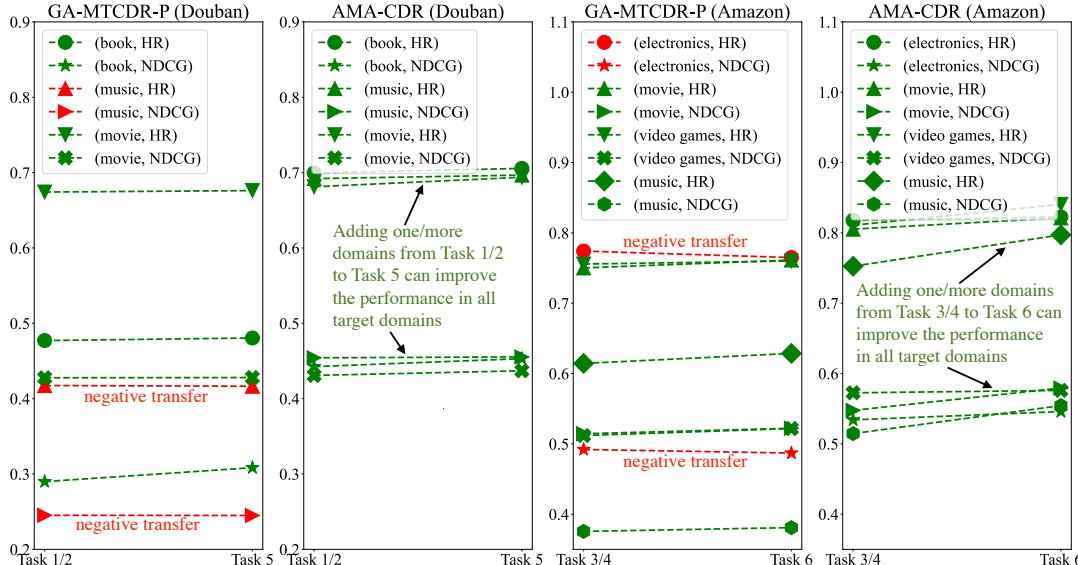

**Figure 5: The experimental results of negative transfer on Douban and Amazon datasets. Note that we add a new domain from Task 1/2 (Book+Movie or Music+Movie) to Task 5 (Book+Music+Movie) and add two new domains from Task 3/4 (Music+Electronics or Movie+VideoGames) to Task 6 (Music+Electronics+Movie+VideoGames).**

## 4 RELATED WORK

### 4.1 Cross-Domain Recommendation

According to transfer paradigm, we classify the existing cross-domain recommendation (CDR) approaches into the following four groups. (1) **One-to-one paradigm**: Most traditional CDR approaches belong to the one-to-one paradigm, which tends to leverage auxiliary information from the source domain (**one**) to improve the recommendation accuracy in the target domain (**one**) (also called as single-target CDR in the literature [66]). These approaches either leverage content information to link and share information across domains [2, 17, 43, 45, 47, 48, 50, 58] or leverage common entities (users/items) as a bridge to transfer their embeddings [5, 13, 15, 18, 23, 29, 34, 35, 52, 61, 64, 68] or rating patterns [10, 56] across domains. (2) **Many-to-one paradigm**: This category of CDR approaches tends to leverage the auxiliary information from multiple domains (**many**) to improve the recommendation accuracy in the target domain (**one**) [22, 60]. (3) **One-to-many paradigm**: There are few related approaches in this category [20, 22, 46], which mainly focus on **one**-dense and **many**-sparse CDR scenarios. (4) **Many-to-many paradigm**: Recently, some CDR approaches have attempted to leverage the rich information from multiple domains (**many**) to enhance the recommendation performance in these multiple domains (**many**) [8, 9, 14, 16, 22, 33, 37, 37, 55, 67]. Compared with the first three paradigms, the many-to-many paradigm is more meaningful because it can almost serve all CDR scenarios.

### 4.2 Avoiding Negative Transfer in MTL & TL

In the literature, some multi-task learning (MTL) approaches [31, 32] and transfer learning (TL) approaches [6, 42, 53] have attempted to address this problem by measuring the distribution gap between two domains or leveraging adversarial learning to re-weight the importance of samples, and thus they can generate a kind of hidden

irrelevance among different domains to avoid the negative transfer. However, some transfer-related prior knowledge, i.e., data density, and the posterior knowledge, i.e., sample uncertainty, in the area of CDR are ignored in these approaches.

### 4.3 Attention and Active Learning for CDR

Because we employ attention and active learning for our proposed AMA-CDR, we also review the related literature as follows. (1) **Attention**: Attention is first introduced in [1], which provides a more accurate alignment for each position in a machine translation task. In the area of CDR, the attention mechanism is applied to select more informative parts of the auxiliary domains to help the target domain [25, 26, 65]. (2) **Active learning**: The key idea of active learning is that machine learning algorithms can achieve greater accuracy with fewer training samples if the algorithms can actively choose the samples from which are learned [44]. As for CDR, active learning can actively choose the uncertain samples that urgently need auxiliary information from other domains [38, 60].

## 5 CONCLUSION AND FUTURE WORK

In this paper, we have proposed a novel **A**ctive **M**asked **A**ttention framework, i.e., AMA-CDR, for many-to-many CDR scenarios. Specifically, we have designed an end-to-end graph framework to reduce the objective distortion between graph embedding and transfer strategy. More importantly, we have proposed an active mask to address the problem of negative transfer. Also, we have conducted extensive experiments to demonstrate the superior performance of our AMA-CDR and verify that the recommendation performance in the target domain can be close to or even overhauled in the source domain. In the future, we plan to study more prior knowledge and posterior knowledge to control the demand probability of transferred information and choose suitable uncertain samples.

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
