# OpenReview forum: "An Active Masked Attention Framework for Many-to-Many Cross-Domain Recommendations"
_acmmm.org/ACMMM/2024/Conference — MM2024 Poster_

### Official Review · Reviewer_8NwB · 2024-05-20

**Rating:** 3
**Confidence:** 3

**Summary:**

This article proposes active masked attention in the problem setting of multi-domain Graph-based CDR to overcome the negative transfer issue, thereby improving the performance of CDR and achieving state-of-the-art performance on several datasets. Overall, this paper selects a BITGCF-like model as the baseline and adds an attention module to improve performance.

**Strengths:**

This paper has the following strengths:
--- 1. The experiments are solid and achieved significant performance improvements.
--- 2. The methodology section is easy to follow.

**Limitations:**

This paper has the following limitations:

--- 1. The writing in the first chapter of the article is unclear and difficult to follow. For instance, in line 137, it mentions "there are two main problems in the literature," but which two problems does it refer to? Is one of the problems negative transfer? What is the other problem?
--- 2. The paper introduces a new metric, "New Goal," but it seems to me that the paper does not clearly define what "the new goal" is. My question is: in Table 3, under what circumstances can a recommended task be considered to have met the criteria for "achieving the new goal"?
--- 3. In my opinion, there is no strict logical relationship between "whether the sparser domain can outperform the denser domain" and "practically usable level." It would be better if the author provided a rigorous discussion on these issues.
--- 4. The apperance of the paper needs to be improved.

**Suitability:**

2

---

### Official Review · Reviewer_oCWk · 2024-05-24

**Rating:** 4
**Confidence:** 4

**Summary:**

This work focuses on many-to-many cross-domain recommendation scenarios and proposes a novel active masked attention framework, i.e., AMA-CDR. AMA-CDR adopts an end-to-end graph embedding to reduce the objective distortion between graph embedding and embedding combination and leverages an active mask for the embedding combination to ease negative transfer. The experiments show that their proposed model outperforms other baselines and achieves the new goal.

**Strengths:**

1. This work mainly focuses on the hardest but meaningful CDR scenario, i.e., many-to-many CDR, and improves the recommendation performance in the sparse domain to achieve a practically usable level, with a clear and well-founded motivation.
2. The proposed model and the two designed modules, i.e., an end-to-end graph embedding and an active mask, by the authors are reasonable and highly innovative, providing a complementary approach to existing methods.
3. The authors conducted extensive experiments to demonstrate the superiority of the proposed AMA-CDR framework.
4. This work exhibits a well-organized and logically sound structure, with clear and easily understandable writing.

**Limitations:**

1. In the introduction section, regarding the research problem (line 139), the description of the objective distortion problem is missing.
2. This work is not open source and is not easy to follow.
3. In the experimental section, for the many-to-many CDR task, the number of domains is at most 4, and more domains need to be validated.
4. In section 3.1.2, the user/item embedding dimension is set to 32. However, in more existing research work, the embedding dimension is set to either 64 or 128. Further clarification from the authors on this parameter setting would be appreciated.
5. In section 3.1.3, the authors state that the experiment conducts the top-N recommendation, but in Table 3 only top-10 results are given. Then $N$ in 3.1.3 should be changed to 10.

**Suitability:**

3

---

### Official Review · Reviewer_EsKY · 2024-05-26

**Rating:** 4
**Confidence:** 1

**Summary:**

The paper  presents a novel approach to address the challenges faced in cross-domain recommendation systems (CDR). The authors identify the problem of negative transfer, which occurs when the knowledge transferred from source domains to a target domain results in reduced recommendation accuracy, particularly in sparse domains. To tackle this, they propose the Active Masked Attention framework (AMA-CDR) for many-to-many CDR scenarios.

**Strengths:**

1.The paper is well-structured, with a clear problem statement, methodology, and evaluation.
2.The proposed AMA-CDR framework is innovative and addresses a significant challenge in CDR systems.
3.The results are promising, demonstrating the effectiveness of the model.

**Limitations:**

1.the paper would benefit from a more detailed discussion on how the active mask is applied in practice and any potential limitations or challenges in implementing the framework in real-world scenarios.
2.Additionally, future work could explore how the framework scales with an increasing number of domains and how it performs in dynamic environments where domains are continually evolving.

**Suitability:**

2

---

### Official Review · Reviewer_7nRu · 2024-05-26

**Rating:** 5
**Confidence:** 3

**Summary:**

The paper addresses the common issue of data sparsity in cross-domain recommendation systems and then proposes a model, AMA-CDR, which designs an end-to-end graph framework to reduce the objective distortion between graph embedding and transmission strategy and introduces an active masking approach to tackle the problem of negative transfer. Experiments were conducted on Douban and Amazon datasets, demonstrating superior performance compared to baseline models and alleviating the data sparsity issue to some extent.

**Strengths:**

1. The questions addressed in this article are very novel and the ones proposed are quite creative.
2. The structure diagram of the method in this article is very clear, making it easy for people to understand the author's intention.
3. The experiment was conducted relatively comprehensively, with a wide range of dimensions for comparison.

**Limitations:**

1. When describing the shortcomings of other models, the author mentions "is still too low in real application scenarios". Is there any data to support this point? What is the real application scenario referred to?
2. The author's final experimental results indicate that using this method can make the recommendation performance on sparse data exceed that of a single rich domain without using cross-domain methods. This result is quite surprising because usually the sparsity of the data largely determines the quality of the results.
3. The author addressed two issues but only mentioned one in the introduction section.

**Suitability:**

2

---

### Meta-Review · Area_Chair_BeSE · 2024-07-04

**Recommendation:** Accept (Poster)
**Confidence:** 2

**Metareview:**

This paper propose a novel Active Masked Attention framework, i.e., AMA-CDR, for many-to-many CDR scenarios which designs an end-to-end graph framework to reduce the objective distortion between graph embedding and transmission strategy and introduces an active masking approach to tackle the problem of negative transfer. Experiments were conducted on public datasets, demonstrating superior performance compared to baseline models and alleviating the data sparsity issue to some extent.

The paper is well-written and illustrate the proposed method clear. But there are one concern raised by reviewers have not been addressed in the rebuttal materials:
1) The author's final experimental results indicate that using this method can make the recommendation performance on sparse data exceed that of a single rich domain without using cross-domain methods. This result is quite surprising because usually the sparsity of the data largely determines the quality of the results.

Overall, the concerns are well addressed by the authors. The proposed method addressed an important problem in recommendation area and the contribution is sufficient. I would recommend to accept the paper based on the discussions between reviewers. But I am an expert on the recommendation area.